# The ecological origins of snakes as revealed by skull evolution

Filipe O. Da Silva[1], Anne-Claire Fabre[2], Yoland Savriama[1], Joni Ollonen[1], Kristin Mahlow[3], Anthony Herrel[2], Johannes Müller[3] & Nicolas Di-Poï [1]

The ecological origin of snakes remains amongst the most controversial topics in evolution, with three competing hypotheses: fossorial; marine; or terrestrial. Here we use a geometric morphometric approach integrating ecological, phylogenetic, paleontological, and developmental data for building models of skull shape and size evolution and developmental rate changes in squamates. Our large-scale data reveal that whereas the most recent common ancestor of crown snakes had a small skull with a shape undeniably adapted for fossoriality, all snakes plus their sister group derive from a surface-terrestrial form with non-fossorial behavior, thus redirecting the debate toward an underexplored evolutionary scenario. Our comprehensive heterochrony analyses further indicate that snakes later evolved novel craniofacial specializations through global acceleration of skull development. These results highlight the importance of the interplay between natural selection and developmental processes in snake origin and diversification, leading first to invasion of a new habitat and then to subsequent ecological radiations.

[1] Program in Developmental Biology, Institute of Biotechnology, University of Helsinki, 00014 Helsinki, Finland. [2] Département Adaptations du vivant, UMR 7179 C.N.R.S/M.N.H.N., 75231 Paris Cedex 5, France. [3] Museum für Naturkunde, Leibniz Institute for Evolution and Biodiversity Science, 10115 Berlin, Germany. Correspondence and requests for materials should be addressed to N.D.-P. (email: nicolas.di-poi@helsinki.fi)

Acentury of anatomical and phylogenetic studies have established that snakes evolved from lizards[1,2], these two groups forming together one of the most-specious clades of terrestrial vertebrates—the squamate reptiles. In addition, the evolution and development of the limb and axial skeleton have been recently assessed in snakes, suggesting a correlation between limb loss and body elongation[3–8]. However, studies assessing the early ecological and evolutionary origins of snakes have so far largely focused on discrete morphological differences, and the adaptive role of skull shape development in the origin and diversification of snakes remains to be tested at large scale. The interplay between development and natural selection in driving morphological skull disparities associated with the lizard-to-snake transition remains poorly understood. Therefore, in this study, we performed a large-scale and integrative geometric morphometric analysis of skull bones across squamates to help clarify the ecological and evolutionary origins of snakes.

Conflicting ecological hypotheses for early snakes, including aquatic[9–11], terrestrial[12–14], fossorial[15–18], or even multiple habitats[19], have been proposed based on cladistic analysis of discrete traits. Central to this debate are the paucity of intact well-preserved snake fossils[16–19], the difficulty of deciphering squamate phylogenetics[20–24], sampling variability and incompleteness[8–11,13–26], and incompatible morphological character coding when convergence is expected[9,10,12–20,26]. In addition, ancestral ecologies have been typically hypothesized based on sister clades of snakes only, without formal ancestral character state estimates[8–20,25,26]. Recently, modern phylogenetic comparative methods have been developed to estimate the ecological state of snake ancestors[23], but with a limited taxon sampling that hampered the phylogenetic signal. Similarly, investigations of the evolution of discrete ossification sequences in skull development could not discern between different ancestral ecological scenarios[27].

The analysis of morphological data using complementary geometric morphometric approaches has the potential to shed light on these issues[28]. In support of that, the comparison of the inner-ear shape in a limited number of snake species has already been used in the context of snake origins[29,30], and recent geometric morphometric studies integrating ecological and/or developmental data revealed new insights into skull evolutionary specializations in several lizard and snake radiations[31–34]. The diversity in cranial structure of squamates is remarkable and appears tightly linked to functional and constructional demands within specific clades of lizards or snakes[34–39], suggesting that large-scale comparisons of skull shape and size across the whole of Squamata could offer a holistic framework to address the ecological origin of snakes. Equally, cranial shape associated with ontogeny and heterochronic processes—changes in the timing and/or rate of developmental events—have been implicated in cranial evolution at different taxonomic levels in squamates[27,31,40–42] and other major vertebrate lineages such as mammals and archosaurs[43,44]. Thus far, studies tackling the evolutionary origin of snakes have largely ignored morphometric and ontogenetic information, as well as the importance of developmental mechanisms for understanding the ecological origins of snakes.

Here we performed a large-scale and integrative characterization of skull shape evolution in squamates, by covering all major groups of lizards and snakes, using a geometric morphometric approach integrating developmental, embryological, ecological, phylogenetic, and paleontological data. We hypothesized that skull shape and size are linked to particular habitat features; hence, ancestral snake ecologies could be inferred from skull shape parameters. In complement, and to formally test the recently revived hypothesis that snake skulls evolved by heterochrony[27,40,45], we quantified ontogenetic skull shape variation by analyzing multivariate ontogenetic trajectories in a unique data set of squamate embryos. Strikingly, our data reveal that while the most recent common ancestor (MRCA) of crown snakes had a small skull with a shape fully adapted to a fossorial lifestyle, all snakes plus their sister group evolved from a terrestrial form with non-fossorial or non-leaf-litter behaviors, thus indicating a surface-terrestrial-to-fossorial scenario at the origin of snakes. In addition, we demonstrate that the unique skull features of modern snakes later evolved by peramorphosis through global acceleration in the rate of craniofacial development during embryogenesis. This set of results demonstrates the importance of the relationships between skull form, function, and development in the major ecological radiations of snakes to different habitats, and provides a new framework to understand the origin and evolutionary history of snakes.

## Results

**Exceptional skull shape variation in squamates.** Our principal component analysis (PCA) performed on Procrustes coordinates of skulls from a large data set of adult squamate species (302 and 91 species for two-dimensional (2D) and three-dimensional (3D) data, respectively; Fig. 1, Supplementary Fig. 1, and Supplementary Note 1) generated a morphospace defined by two principal components, PC1 and PC2, which together account for more than 60% of the total shape variation in both 2D and 3D analyses (Fig. 2a and Supplementary Figs 3 and 4). Importantly, these two PCs provide the best approximation for the total skull shape variance, as other subsequent PCs explain <10% of the total variation, and our systematic comparisons of 2D and 3D data confidently indicated similar skull shape changes between and within lineages of lizards and snakes (see below and Supplementary Notes 1 and 2). Remarkably, as reflected by the separated distribution of most snake and lizard species at negative and positive values, respectively, PC1 clearly distinguishes snakes from lizards (Fig. 2a); the only exceptions to this pattern are putative convergent morphologies located between positive PC1 and negative PC2 values, which include specimens from independent lineages such as scolecophidians (all families), alethinophidians (Uropeltidae, Aniliidae, Anomochilidae, and Cylindrophidae families), and legless lizards from all major lineages. Wireframes displaying skull shape variations along the two main PC axes further indicate that PC1 positive values are associated with both a more flattened snout and an expanded skull roof (Fig. 2b, left panel), as exemplified by the formation of a prominent crest on the parietal bone at extreme values (Fig. 2a). A mirrored pattern of skull shape is observed toward negative PC1 values, where the braincase is dorsally compressed and the snout enlarged, accentuating a cylindrical shape typical of modern snakes (Figs. 1k–m and 2b). Interestingly, lizards display greater shape variation along the PC2 axis, with most species being scattered along positive values that characterize a triangular skull shape (Figs. 1b, e, f and 2a) with an enlarged ocular region, a tall braincase, a highly reduced parietal wall, and a robust snout derived from the expansion of both maxillary and nasal bones (Fig. 2b, right panel). At negative PC2 values, the parietal region completely encases the midbrain with a lateral downgrowth, and several modifications such as the compression of the ocular region and the strong posterior extension of both braincase and pre-maxilla regions make the general skull shape more cylindrical (Figs. 1d, g, j and 2b). Notably, the latter morphological changes are typical characteristics of the putative convergent forms described above. Other major skull shape changes present both in lizards and snakes include the shortened quadrate bone along its dorso-ventral axis at negative PC2 values, a condition also found

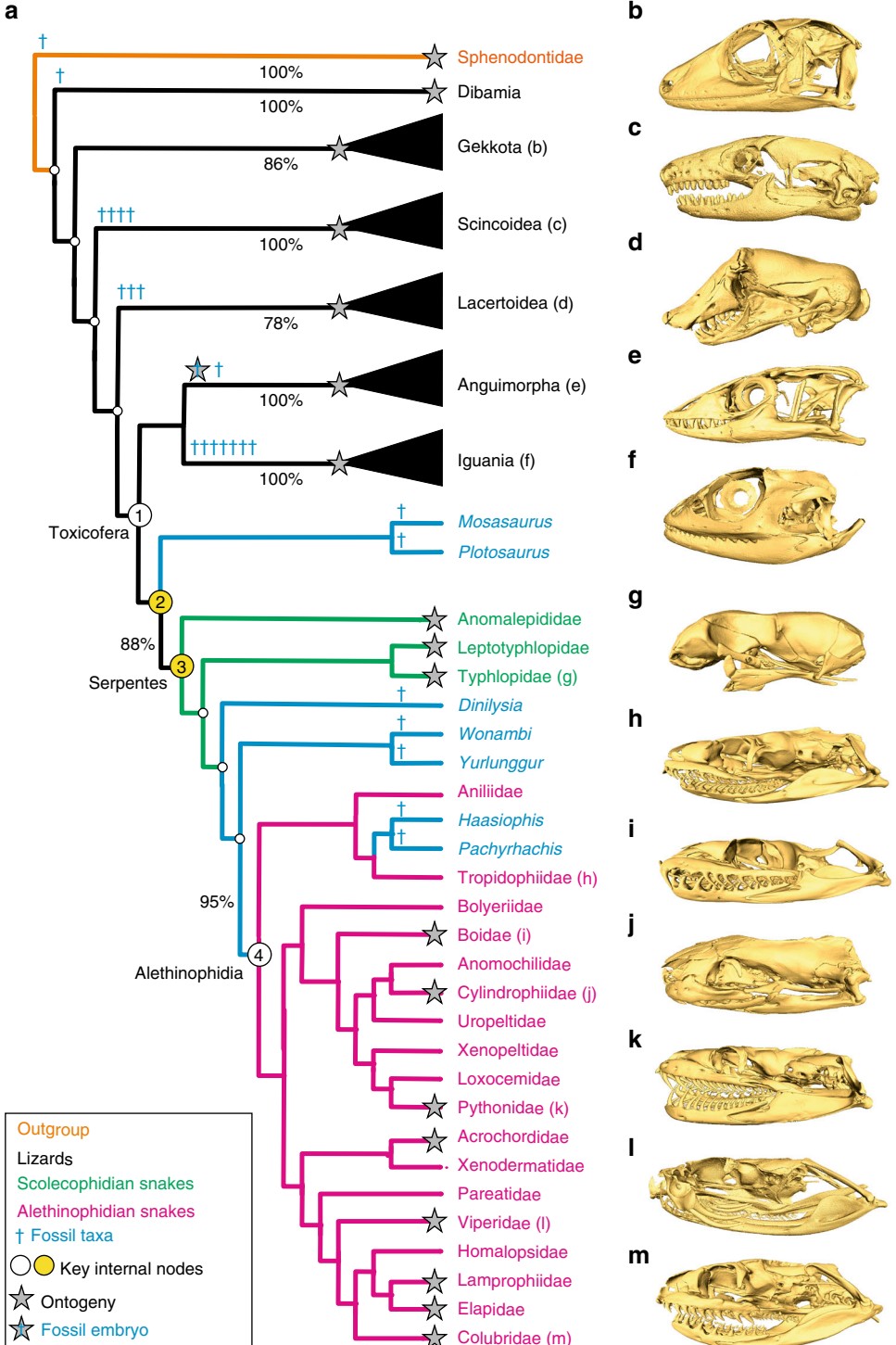

**Fig. 1** Phylogeny and skull diversity of squamates. **a** Simplified phylogenetic hypothesis used in this study, adapted from the most inclusive and recent molecular[21, 64] as well as combined molecular and morphological phylogenetic studies on squamate evolution[22–24], and rooted using Sphenodontidae (tuatara). Major extant squamate lineages (lizards, black color; scolecophidians, green; alethinophidians, red), tuatara (orange), and fossil genera (blue) are indicated by the same color code throughout the entire manuscript; similarly, the same number code is used for all key internal nodes relative to the origin and diversification of snakes (1, MRCA of Toxicofera; 2, MRCA of snakes and their sister group; 3, MRCA of crown snakes; 4, MRCA of Alethinophidia). The percent of extant families collected at a particular branch is shown for major lineages. The positions of sampled fossils and ontogenies are indicated by blue crosses and stars, respectively. (**b**–**m**) 3D-rendered adult skulls of selected representative squamates at indicated position in the phylogenetic tree: *Aeluroscalabotes* (**b**), *Acontias* (**c**), *Leposternon* (**d**), *Varanus* (**e**), *Iguana* (**f**), *Letheobia* (**g**), *Tropidophis* (**h**), *Boa* (**i**), *Cylindrophis* (**j**), *Python* (**k**), *Bitis* (**l**), *Boaedon* (**m**)

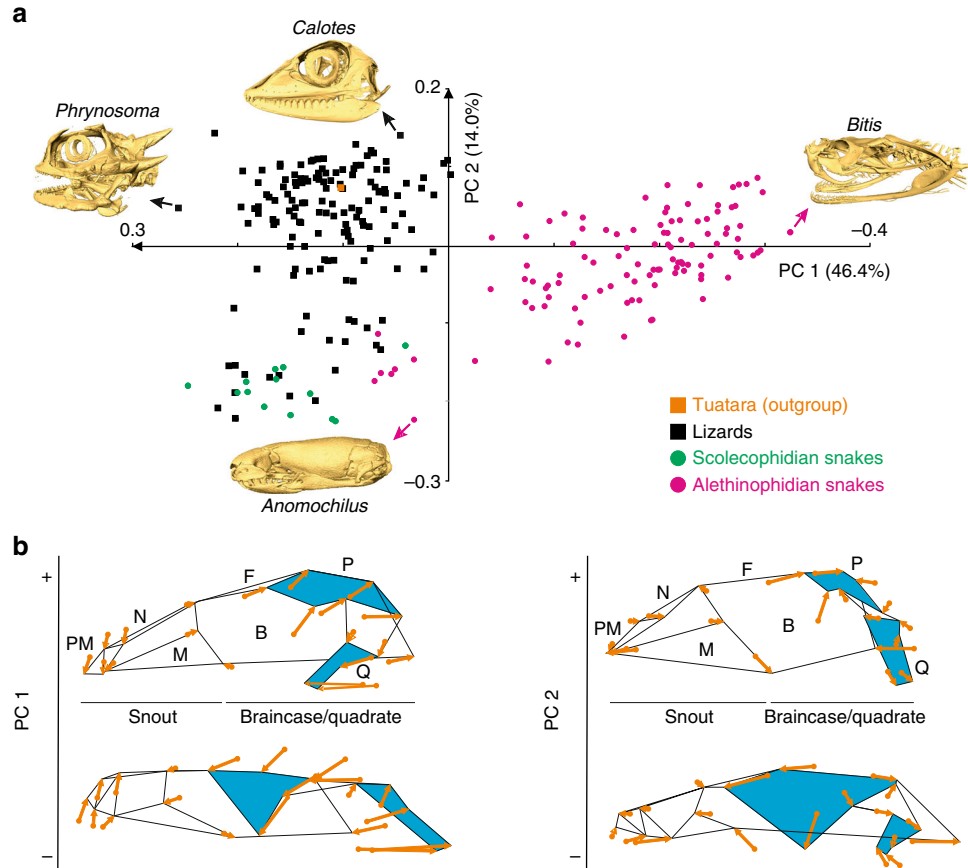

**Fig. 2** Principal component analysis of skull shape variation in extant adult squamate species. **a** Plot of principal component (PC) scores showing the skull shape distribution of lizards (black squares), scolecophidians (green circles), alethinophidians (red circles), and tuatara (outgroup, orange square) for which 2D data were available. Numbers in brackets indicate the percentage of variance explained by each of the PC axes. The 3D-rendered skulls (with species names) corresponding to the observed extreme species in both positive and negative directions are shown. **b** For each PC, the extreme shapes at positive (+) and negative (−) values are depicted as skull outlines with vectors of landmark displacement (orange arrows) indicating the direction and intensity (reflected by vector length) of shape changes from the mean shape. Different regions of the skull are visible by black lines connecting landmark points: B, back of the skull; F, frontal; M, maxilla; N, nasal; P, parietal; PM, pre-maxilla; Q, quadrate. Parietal and quadrate bones, showing the greatest variations in size and shape, are shaded in blue color

in convergent morphologies (Figs. 1d, g, j and 2b), and the different curvature and projection of the quadrate ventral articular surface along PC1 values (Fig. 1b, f for lizards and Fig. 1g, j for snakes).

**Reconstruction of ancestral snake skull shapes.** To investigate the morphological transitions in patterns of phylomorphospace occupation, we estimated the evolution of skull shape with both unweighted and weighted squared-change parsimony algorithms[46] in a phylogenetic context, using a main topological hypothesis integrating fossil data and compiled from the most recent and inclusive molecular as well as combined molecular and morphological phylogenetic studies on squamate evolution (Fig. 1, Supplementary Fig. 1, and Supplementary Notes 1 and 2). Importantly, several alternative composite hypotheses as well as different recent combined molecular and morphological phylogenetic trees[22–24] were also tested to include different species numbers and/or branch length information but also to address the phylogenetic uncertainty of some fossils such as Dinilysia[22–24] and mosasauroids[22,24] (Supplementary Notes 1 and 2). This phylomorphospace approach clearly shows phylogenetic trends, as indicated by the minimal overlap of branches from major lineages, and suggests the convergent evolution of several snake

and lizard species from different families toward positive PC1 and negative PC2 values (Fig. 3). As expected from these observations, a significant phylogenetic signal was identified using a multivariate $K$-statistic[47] ($K$-value = 0.53; $p$-value = 0.001; Supplementary Note 2), but the relatively low $K$-value indirectly indicates that the phylogenetic relatedness is not the only factor affecting shape evolution, thereby further implying the existence of convergent shape patterns (Fig. 3 and Supplementary Fig. 3). Interestingly, in all tested phylogenetic hypotheses, the positions of our reconstructed MRCAs of Toxicofera (Figs. 1 and 3, node 1), snakes and their sister group (Figs. 1 and 3, node 2), and crown snakes (Figs. 1 and 3, node 3) indicate a lizard-to-snake transition proceeding along the PC2 axis (see also Supplementary Fig. 3 and Supplementary Notes 1 and 2), thus reflecting the importance of increased skull encasing and cylindrical skull shape in early snake evolution (Fig. 2). Unexpectedly, however, none of the tested well-preserved Cretaceous skull fossils, including marine (*Pachyrhachis* and *Haasiophis*)[9,15], terrestrial (*Yurlunggur* and *Wonambi*)[12,14], or terrestrial/burrowing (*Dinilysia*)[26,29] species were found near these estimated ancestral skull shapes (Fig. 3, Supplementary Fig. 4, and Supplementary Note 2), indicating that they are not fully representative of the early snake skull shape. Instead, as also recently proposed by others[15–17,30], such fossils likely represent specialized evolutionary offshoots of the initial

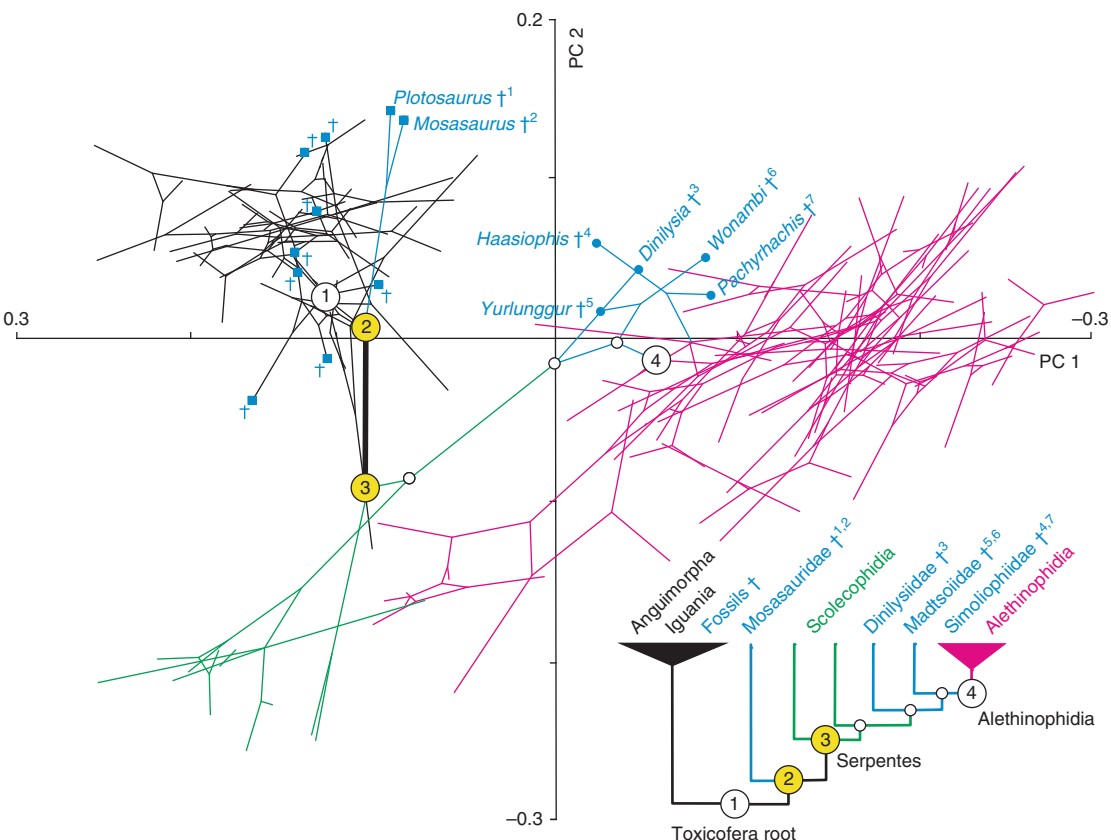

**Fig. 3** Origin and diversification of skull morphology in snakes. Phylomorphospace created by projecting the squamate phylogeny presented in Fig. 1 onto the morphospace delimited by the first two PC axes. For clarity, only Toxicofera specimens are shown. Ancestral character estimation for each internal node was performed using squared-change parsimony, and branch lengths were derived from estimated divergence times. The position of lizard and snake fossils (with species names indicated for specimens already proposed to be at the origin of snakes) is shown by blue squares and blue dots, respectively, and key ancestral nodes are numbered and colored as in Fig. 1. The estimated lizard-to-snake transition took place between nodes 2 and 3 (yellow nodes), as indicated by the thick bold line. The bottom right cladogram represents a simplified phylogenetic tree of the major Toxicofera lineages (including fossil families, with colors as in Fig. 1) and the ancestral nodes shown in the phylomorphospace

snake diversification. Similarly, extinct mosasaur and extant varanid lizard species (Supplementary Note 2), which are both classically recovered or a priori assigned as the ancestral lineage from which snakes evolved[9,10,23], are located distant from the estimated MRCAs. Finally, our phylomorphospace analysis indicates that scolecophidians radiated from a less-specialized skull shape (Fig. 3 and Supplementary Fig. 3), as also supported by recent morphological and genomics data[17,29,48]. This indicates that these snakes are also not the best representative of the ancestral skull condition, even though closer to it than any other tested species. Hence, the ancestral skull shape of snakes cannot be directly inferred from the shape found in the aforementioned extant and extinct lineages.

**Ecological origins of snakes**. Skull morphology is well known to be affected by different selective pressures such as feeding performance, diet, and behavior[34,38,39], but habitat specializations are also expected to be of major influence[36]. To assess the ecological origin of snakes, we explored potential relationships between skull shape and habitat preference, by plotting specific ecologies upon shape distribution in the phylomorphospace. Interestingly, our data indicate cranial shape differences among ecological groups (Fig. 4a and Supplementary Fig. 5) and a significant influence of ecology on skull shapes even after correcting for phylogeny (Supplementary Note 2). Notably, a discrete area

containing all fossorial lizard and snake species clearly separates from other ecologies at negative PC2 values (Fig. 4a and Supplementary Fig. 5), and distance-based convergence measures from Stayton[49] support the significant cranial convergence of such fossorial taxa (Supplementary Table 5 and Supplementary Note 2). In addition, both post hoc pairwise comparisons and discriminant function analysis (DFA) confirmed significant shape differences between some ecological groups, in particular between the fossorial ecology and all other habitat modes (Supplementary Note 2), indicating that shape parameters can be confidently used to estimate ancestral ecologies. Based on the positioning of the reconstructed MRCA of crown snakes and MRCA of snakes and their sister group inside and outside the fossorial cluster, respectively, we hypothesized that the two MRCAs would have a different habitat mode (Fig. 4a). Effectively, using linear DFA and a cross-validation procedure to predict ancestral ecologies (Supplementary Note 2), we recovered with high confidence the fossorial ecology (82% of likelihood) of the MRCA of crown snakes and the non-fossorial (>99.9%) but terrestrial (70%) origin of the MRCA of snakes and their sister group (Supplementary Table 6). To better visualize cranial morphologies associated with both MRCAs, the closest species in multidimensional shape space were warped toward each corresponding ancestral node using the thin-plate spline (TPS) method (Fig. 4b). Consistent with a surface-terrestrial-to-fossorial transition, the MRCA of crown snakes shows a more pronounced cylindrical skull shape with

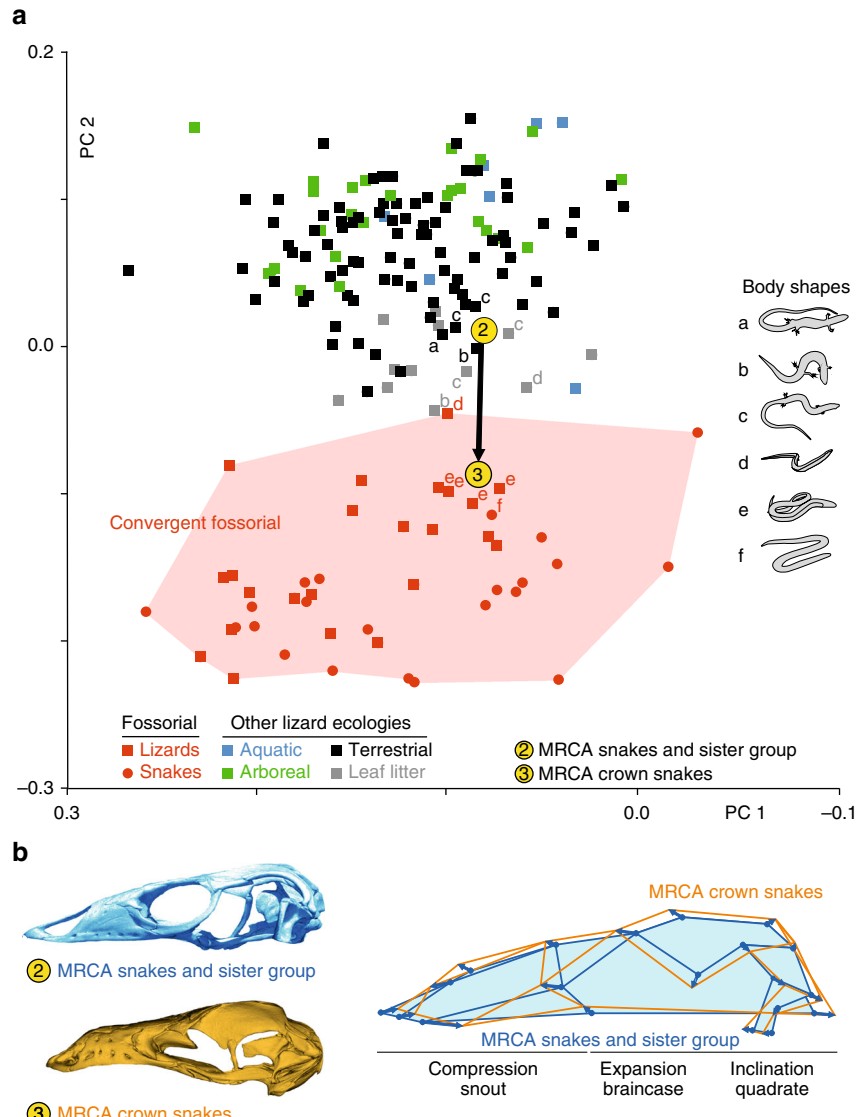

**Fig. 4** Ecological analysis of squamate skull evolution. **a** The habitat preferences of squamates were plotted onto the phylomorphospace presented in Fig. 3. For clarity, branches of the phylogenetic tree are absent, and only lizard (squares) and snake (circles) species located near the reconstructed MRCA of snakes and their sister group and MRCA of crown snakes (yellow nodes 2 and 3, as in Fig. 1) are shown. The thick bold arrow indicates the estimated lizard-to-snake transition. In contrast to other figures, the color code reflects here different ecologies. Red shading indicates the region occupied by convergent fossorial snake and lizard species. Body shapes of species near the reconstructed MRCAs are indicated by letters (see legend in right panel with schematic shapes: a, typical lizard-like body; b, reduced forelimbs; c, reduced forelimbs and elongated body; d, no forelimbs, reduced hindlimbs, and elongated body; e, no limbs, elongated body; f, legless snake-like body). **b** Warped skull surfaces of reconstructed MRCAs of snakes and their sister group (blue color, upper left panel) and crown snakes (orange, bottom left panel), and superimposed skull outlines highlighting shape differences between these two MRCAs (right panel); blue vectors reflect the direction and magnitude of landmark displacement, and major skull shape changes are indicated below the corresponding skull regions

characteristics typical of fossorial species (Fig. 2), including lateral expansions of the parietal region, a posterior expansion of the braincase, as well as a reduced and curved quadrate bone. Similarly, analysis of the general body shape of species near the reconstructed MRCA of crown snakes strongly suggests that several conserved post-cranial modifications associated with life underground in squamates, including body elongation and limb loss, most likely accompanied this transition (Fig. 4a). Comparative analyses, including reconstructions of both cranial and axial skeleton shape would be needed to confirm this trend.

**Evolution of skull size in squamates**. To both assess size diversification within squamates and estimate the ancestral size of

early snake skulls, we estimated centroid size evolution as a proxy for total skull size using squared-change parsimony algorithms in a similar way as for shape analysis. As shown in Fig. 5 and Supplementary Fig. 6, squamate species vary greatly in skull size, but the MRCA of crown snakes was systematically predicted to be smaller than both MRCA of snakes and their sister group and MRCA of alethinophidians in all our analyses (Supplementary Note 2), which is in agreement with its predicted fossorial lifestyle. Interestingly, however, this MRCA was also found to be larger than any tested scolecophidian species, which have likely undergone further miniaturization (Fig. 5, Supplementary Fig. 6, and Supplementary Tables 7 and 8). Importantly, these findings may open the possibility of size effects on shape variation (known as allometric effects), especially for the predicted small fossorial

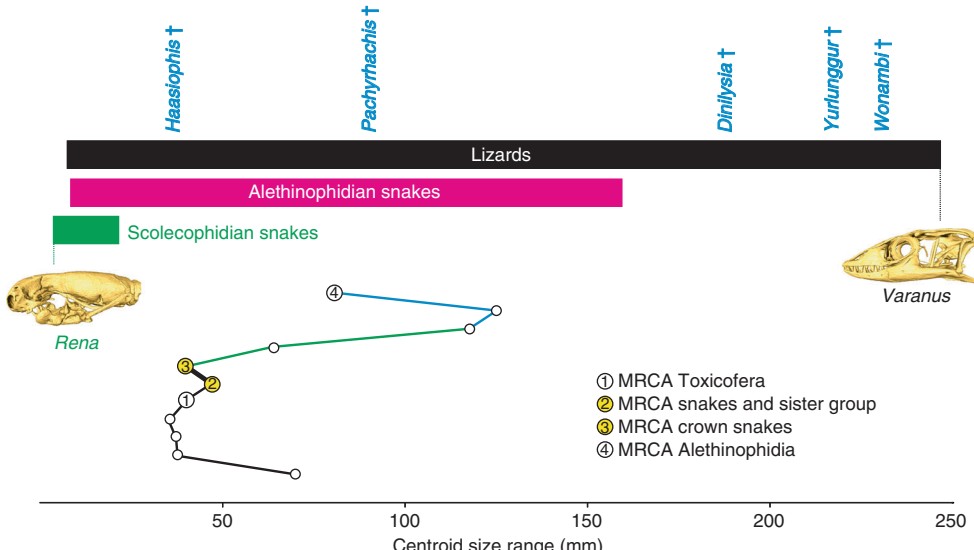

**Fig. 5** Origin and diversification of skull size in snakes. Centroid size variation of skulls for both extant and fossil species were mapped onto the phylogeny presented in Fig. 1, and ancestral estimation for each internal node was performed using squared-change parsimony. For clarity, mosasaur genera are not shown because of their gigantic size, and only phylogenetic tree branches connecting major internal nodes (as in Fig. 1) are drawn. The centroid size range of lizards, scolecophidians, and alethinophidians, as well as the exact position of fossils are indicated on top as reference (with color code as in Fig. 1). 3D-rendered adult skulls of extant species with extreme centroid size values (*Rena* and *Varanus*, position indicated with colored lines) are shown

ancestry of crown snakes. To explore this aspect, we tested the influence of allometry by means of multivariate regressions (Supplementary Note 2) and found significant correlations between size and shape in both our 2D and 3D data sets ($p$-values < 0.0001). However, only a small fraction of total shape variation could be accounted for by allometry (about 6% in our comprehensive 2D data set, Supplementary Table 9), thus indicating that allometric corrections should have limited effects on the total inference of shape patterns. This was indeed confirmed by removing the effects of allometry within morphospace analysis, which almost exclusively affects the skull shape distribution of some scolecophidian species along the PC2 axis (Supplementary Fig. 7 and Supplementary Note 2), without altering the general pattern of other lizard and snake species. In particular, other fossorial species like amphisbaenian lizards show a strikingly conserved pattern of shape distribution after allometric correction (Supplementary Fig. 7), and no significant allometry was observed when this group was treated separately ($p$-value < 0.001), in contrast to scolecophidians ($p$-value = 0.0189). This argues for an independent origin of fossoriality between amphisbaenians and snakes, as also supported by recent phylogenetic studies[21,22,24,50].

Altogether, these results indicate that allometry was presumably critical in the early evolution of snakes toward a fossorial behavior, and suggest that allometry played a different role in the evolution of fossoriality among squamates. In addition, our predicted size for the MRCA of crown snakes strongly rejects the hypothesis of a large, terrestrial pro-boid snake ancestor[12,14,23], but is in agreement with the range of relatively small skull fragments recovered for Cretaceous fossorial snakes such as *Coniophis*[17], *Najash*[16], and the enigmatic *Tetrapodophis*[18]. Unfortunately, the incomplete or damaged skull available for these latter fossils hampers their inclusion into our ecological analyses.

**Ontogenetic and heterochronic processes in snake evolution**. Among evolutionary developmental processes, heterochrony—defined as an evolutionary change in the timing, duration, and/or

rate of development—is widely regarded as one of the most important evolutionary mechanisms driving morphological evolution in vertebrates[41–44]. To assess both cranial ontogeny and the impact of heterochrony on squamate skull evolution, we used a unique embryonic data set covering 50% of all lizard and snake families (Fig. 1 and Supplementary Fig. 1) to quantify and compare the geometric properties (path length, direction, and angle) of ontogenetic trajectories generated by vectors of shape changes between younger-to-older specimens in PCA morphospace (Supplementary Figs 8 and 9 and Supplementary Note 3). Remarkably, both the average length and angle of ontogenetic shape trajectories remain largely conserved between snakes and lizards (Supplementary Table 10, Supplementary Figs 8 and 9, and Supplementary Note 3), and ontogenetic changes in skull shape are almost exclusively directed toward negative PC1 values (Supplementary Fig. 8), where alethinophidian snakes are located. The ontogenetic morphospace indicates that these conserved trajectories are particularly linked to phylogenetic transitions in the shape of the quadrate bone, including shaft elongation, curvature loss, and caudal projection, but also to other skull features such as shortening of the snout. Interestingly, such shape changes are believed to have contributed to the evolution of more flexible skulls with a wider gape size within alethinophidians[2], further suggesting that developmental innovations are linked to clade diversification and exploitation of new ecologies and feeding strategies in snakes.

The overall conservation of ontogenetic trajectories allowed us to test for different global heterochronic hypotheses[51]. Two major types of heterochronic processes—paedomorphosis and peramorphosis—have been defined in evolutionary developmental biology, but peramorphosis has been particularly hypothesized to underlie the rise of new morphologies through changes in developmental timing[52], including an extended period (hypermorphosis), an earlier onset (predisplacement), or an increased rate of development (acceleration) of descendent lineages compared to ancestors. The regression of shape changes onto centroid size as a proxy for ontogenetic time, a widely used method to detect heterochronic trends[31,40–42,51,52], clearly

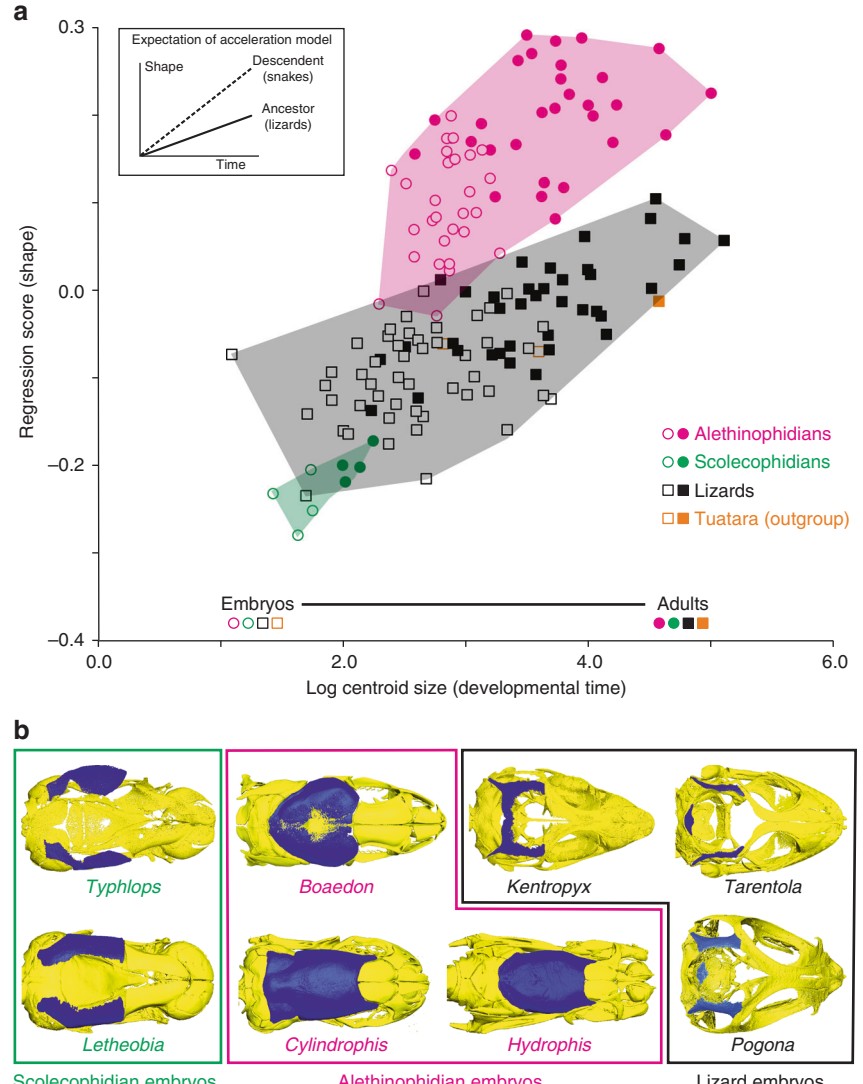

**Fig. 6** Ontogenetic and heterochrony analyses. **a** Regression analysis of 2D shape (regression score) onto log-centroid size for lizard (black squares), scolecophidian (green circles), alethinophidian (red circles), and tuatara (outgroup, orange squares) ontogenies. Embryos are shown as open circles (for snakes) or squares (for lizards and outgroup). The top left inset shows the expected regression lines for an acceleration model of heterochrony. **b** Dorsal views of 3D-rendered skulls from scolecophidian (*Typhlops* and *Lethobia*), alethinophidian (*Boaedon*, *Cylindrophis*, and *Hydrophis*), and lizard (*Kentropyx*, *Tarentola*, and *Pogona*) stage 10 embryos, showing the ossification levels of the different skull bones; delayed ossification of the parietal bone (blue color) in all scolecophidians and lizards is highlighted

demonstrates steeper regression slopes and angles for snakes in comparison to lizards (Fig. 6a, Supplementary Fig. 10, and Supplementary Table 11), thus indicating a peramorphic process through acceleration (Supplementary Note 3). This model was further supported by comparing the total duration of embryonic development in 126 lizard and snake species, which revealed no significant differences between the two groups (Supplementary Tables 12 and 13 and Supplementary Note 3). Also compatible with a global acceleration model, comparisons of the ossification degree of both frontal and parietal bones, the last two bones to complete ossification in squamates[38], systematically show more advanced ossification in alethinophidians than in lizards at equivalent late pre-hatchling embryonic stages (stage 10; Fig. 6b, Supplementary Table 14, and Supplementary Note 3), which is in accordance with the incompletely ossified skulls reported for postnatal stages in several lizard lineages[53]. Interestingly, most of our tested scolecophidian taxa also exhibit a surprisingly poorly ossified skull roof at late embryonic development, or even at

juvenile stages, thus resembling the lizard condition (Fig. 6b, Supplementary Table 14, and Supplementary Note 3). The latter results are contradictory to the recent assumption that the rate of development would be similar between scolecophidians and alethinophidians[27], and rather indicate signs of heterochrony such as paedomorphosis, as also supported by previous postnatal skull inspections[54,55] and the truncated position of scolecophidians in our regression analysis (Fig. 6a).

## Discussion

Our comparative geometric morphometric study demonstrates that the skull shape distribution from lizards to snakes is gradual but does not occupy the entire estimated morphospace, indicating that lizards could not have transitioned to snakes by any other evolutionary path than through fossoriality. In support of this hypothesis, we identified significant convergence for different lizard and snake lineages within the miniaturized fossorial

morphospace, thus highlighting the likely critical importance of fossorial transitions in squamate evolution. This also parallels the repeated independent evolution of body elongation and limblessness in squamate species with a fossorial or burrowing lifestyle[56]. In addition, we show that natural selection frequently drove size and shape evolution toward small, encased, and inflexible skulls adapted to a fossorial environment, in accordance with functional hypotheses favoring a fossorial snake ancestor[15–18,29]. Interestingly, however, the MRCAs of both Toxicofera and snakes and their sister group were reconstructed here as being terrestrial but non-fossorial, thus revealing a new underexplored scenario in the discussion of snake evolution, which is predominantly focused on a marine-to-marine, terrestrial-to-terrestrial, or fossorial-to-fossorial transition from the Late-Jurassic period to Early-Cretaceous[9,12,14–18,23,25]. These findings are consistent with a similar terrestrial-to-fossorial transition associated with the origin of fossoriality in other squamates such as amphisbaenians and Gymnophthalmidae[36,50], and resurrect an early twentieth-century hypothesis claiming that snakes originate from terrestrial lizard organisms[57]. Hence, we believe that the current hypotheses on snake origins could be skewed by incorrect assumptions of similarities between fossorial lizards and snakes, which are, in fact, convergent.

Our phylogenetic comparative developmental data do not favor either the body-first[17] or head-first[19] hypothesis, which propose that head or body evolved through different developmental rates in snakes, respectively. Instead, we show that skull development is accelerated in alethinophidian snakes in comparison to lizards, a process that closely parallels the relatively higher rate of somitogenesis previously reported in snakes during embryogenesis[58], where increased vertebral counts and body elongation are explained by a faster segmentation clock[4,58]. Such developmental similarities rather suggest that skull and body may have evolved jointly in snakes by peramorphosis, through systemic acceleration of developmental mechanisms involved in the differentiation of different parts of the organism, but similar morphometric and ontogenetic analyses of axial anatomy would be needed to confirm this hypothesis. In this context, the reported acceleration of genome evolution in snakes through relaxation of structural constraints in conserved development-related gene-containing regions[59,60] might have facilitated the acceleration of snake development and the origin of new morphologies. Further investigations of the developmental mechanisms underlying skull development in squamates, including the roles of pleiotropic developmental genes such as *Dlx* and *Hox*, might be able to clarify how fast development in snakes evolved. Interestingly, heterochronic processes have already been hypothesized to underlie the reduced connections among palatal, snout, and jaw bones associated with the unique feeding mode in snakes[41,42,45], and changes in the onset of ossification of several skull bones were recently linked to these unique features[27]. These findings are complementary to our observed global heterochronic pattern, and suggest that local or dissociated heterochronic changes might also contribute to patterns of evolutionary changes in the skull of squamates. Finally, our quantitative analysis of squamate skulls complete the scheme on the key role of heterochrony in the rise of new skull morphologies in amniotes, previously reported for archosaurs[43] and mammals[44].

Altogether, this work provides a new framework for the origin and evolutionary history of snakes, which can be refuted or reinforced by future paleontological discoveries and dissection of molecular and developmental mechanisms. Of note, we show that the evolution of the snake skull—feared by some, but intriguing for evolutionists—is a clear example of balance between natural selection (ecology) and temporal regulation of morphogenesis (heterochrony).

## Methods

**Specimen collection**. 3D computed tomography (CT) scans and/or 2D high-quality photographs of skulls were obtained from 326 squamate species (408 skulls), including 279 extant adult or juvenile (298 skulls), 55 embryonic (84 skulls, including 1 fossil embryo), and 23 well-preserved adult fossil (26 skulls) species covering all lineages and most families of squamates (Fig. 1, Supplementary Fig. 1, and Supplementary Tables 1 and 2). The number of extant squamate species sampled represents ~3% of total Squamata, based on the total number of known species reported in the August 15, 2016 version of the Reptile Database (http://www.reptile-database.org). Specimens were primarily sampled from the published literature, the Digital Morphology (DigiMorph) library, reptile colonies at the University of Helsinki (Finland) and Tropicario (Finland), as well as from collections at the Finnish Museum of Natural History (Finland), Museum für Naturkunde in Berlin (Germany), Museum of Comparative Zoology of Harvard University (USA), and American Museum of Natural History (USA). Staging of new unpublished embryos was performed based on the 'standard event system' using external characters[61], and with the help of complete staging tables available for several snake and lizard species[62].

**Geometric morphometrics and multivariate statistics**. Skull shape was extracted by digitizing 61 (for 3D data) and 20 (for 2D) landmarks using Amira 5.5.0 (Visualization Sciences Group) and tpsDig v2.17 software package, respectively (Supplementary Figs 2 and 3 and Supplementary Tables 3 and 4). Standardized skull lateral views were used for 2D data, as they offer a more-inclusive assessment of skull anatomy, after controlling their correct positioning through both the detection of shape outliers in the software MorphoJ v1.06[63] and the comparison of different sources of data (including 2D and 3D skull information) for the same species when available. Bones absent in different squamate lineages such as the temporal bar and supratemporal bones were not included in the analysis. All data were scaled by voxel or pixel size in the respective 3D and 2D package. Landmark coordinates were aligned using Generalized Procrustes Analysis and projected onto tangent space in the package MorphoJ v1.06[63]. The evolutionary patterns of cranial shape disparity were visualized using a PCA as implemented in MorphoJ v1.06. The influence of allometry was tested using a multivariate regression analysis of independent contrasts of shape (Procrustes coordinates) on independent contrasts of size (centroid size), and statistical significance was assessed using a permutation test (10 000 permutations) against the null hypothesis of total independence. Residual scores were also used to correct allometry and verify its relevance to the patterns of shape distribution. TPS technique was used for data interpolation, allowing estimation and visualization of species-specific shape changes through deformation grids with depicted vectors reflecting both the direction and magnitude of changes.

**Analysis of skull shape and size evolution**. To include a large data set of squamate specimens, including extant, fossil, and embryonic taxa, a main composite phylogenetic hypothesis was used by assembling the most recent and inclusive molecular[21,64] as well as combined molecular and morphological topologies[22–24] of squamate evolution (Fig. 1 and Supplementary Fig. 1). For assessing skull shape evolution in a phylogenetic context, a phylomorphospace was generated by first plotting the main PC scores on the phylogenetic tree, and then by reconstructing the ancestral shapes of the internal nodes using both unweighted and weighted (equivalent to maximum likelihood) squared-change parsimony algorithms[46] where ancestral nodes are always estimated intermediate in morphospace. Phylogenetic signal was calculated using a multivariate generalized *K*-statistic[47] in the R-package geomorph v3.0.5[65] available on the CRAN package repository (https://cran.r-project.org/web/packages/), and phylogenetic relatedness was corrected in statistical analyses that require independent observations. To visualize the skull morphologies of reconstructed MRCAs, the closest species in multidimensional shape space (*Calopistes maculatus* and *Acontias meleagris* for MRCA of snakes and their sister group and MRCA of crown snakes, respectively) were warped toward these ancestral nodes using TPS method[66]. Ancestral skull sizes were also reconstructed using squared-change parsimony algorithms and visualized by mapping centroid size changes onto the phylogeny. All ancestral reconstruction calculations were performed in MorphoJ v1.06 using several alternative composite hypotheses but also different recent combined molecular and morphological phylogenies[22–24], including or not fossil data, mosasauroids, and branch length information, as well as with *Dinilysia* and mosasauroids at different positions to address the phylogenetic uncertainty of these fossils[22–24].

**Ecological analysis**. Ecological habitat modes for squamate species were first gathered from the published literature, reptile databases, and/or personal field observations (Supplementary Tables 1 and 2) and then simplified into five main categories: aquatic (including marine and semi-aquatic species); terrestrial (adapted for surface locomotion and foraging, including saxicolous); leaf-litter (terrestrial but living under vegetation layers or surface debris); fossorial (living and foraging underground); and arboreal (adapted for locomotion between tree branches or bushes, including semi-arboreal). To quantify convergent evolution in the different ecological categories, the distance-based convergence measures C1–C4[49] were computed using the R-package convevol v1.1. Significance was assessed in the same

package using 1000 evolutionary simulations along the phylogeny according to a Brownian-motion model[49]. The influence of habitat modes on skull shape (based on the first 11 PCs, accounting for >90% of total shape variation) was initially tested with multivariate analysis of variance (MANOVA) and phylogenetic MANOVA using simulations under a Brownian-motion model using the R-package geiger v2.0.6[67]. The significant separation of different ecologies was then tested using univariate analyses of variance (ANOVAs) followed by post hoc pairwise comparisons as well as DFA, before predicting ancestral ecologies from both allometry-uncorrected and -corrected shape parameters using linear discriminant analysis in the R-package MASS v7.3-47[68].

**Ontogenetic and heterochrony analysis**. The different ontogenetic trajectories for lizards and snakes were generated as vectors of shape changes in both PCA morphospace and regression analyses[69]. The geometric properties of phenotypic trajectories (path length, direction, and angle) between two ontogenetic points (stage 10 embryos and adults) were quantified in snake and lizard species, using the trajectory.analysis function in geomorph v3.0.5[65]. Statistical significance was determined by a random permutation procedure of 1000 iterations. As both the angle and direction of trajectories did not differ in our data set, different heterochronic hypotheses were tested using multivariate regression of shape (Procrustes coordinates) onto log-centroid size as a proxy for developmental time[51]; the slope, length, and angle between descendant trajectories (snakes) in relation to ancestor trajectories (lizards) were compared to predict global heterochronic processes such as modification of the onset/offset, growth rate, or length of development[52]. Variation in the total duration of embryonic development was evaluated in 126 squamate species (only one representative species per genus) with documented incubation or gestation periods, using ANOVA. Finally, the degree of ossification of both parietal and frontal bones was used as a proxy to compare the offset of skull development in 35 different snake and lizard species at fixed pre-hatchling embryonic stage (stage 10[61]).

**Data availability**. 2D photographs and 3D CT data are in part publicly available in the published literature and www.digimorph.org (Supplementary Information). The remnant data are available through the corresponding author, upon reasonable request. Landmarks are available in the Supplementary Information.

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

## Acknowledgements

We thank Ilpo Hanski and Martti Hildén (Finnish Museum of Natural History, Helsinki, Finland), Peter Giere and Frank Tillack (Museum für Naturkunde, Berlin, Germany), José Rosado (Harvard Museum of Comparative Zoology, Cambridge, USA), and David Kizirian (American Museum of Natural History, New York, USA) for specimen loans; Jessie Maisano (University of Texas, USA) for DigiMorph data sharing; Jarmo Lanki (Tropicario Helsinki) for eggs from some snake species; Arto Koistinen (University of Kuopio, Finland) as well as Aki Kallonen and Heikki Suhonen (University of Helsinki, Finland) for access to X-ray Computed Tomography Facilities; Fabien Lafuma, Umair Seemab, Julia Eymann, Pavla Lockerová, and Eveliina Karjalainen for technical assistance; Sylvain Gerber (Muséum National d'Histoire Naturelle, France) for providing R-scripts for image warping; Miriam Zelditch for an updated version of the "convevol" package, as well as Pierre-Henri Fabre, Jukka Jernvall, Jacqueline Moustakas, and Evo-Devo community (University of Helsinki, Finland) for helpful discussions. This work was supported by funds from the Doctoral Program in Wildlife Biology Research (LUOVA) (to F.O.D.S.), University of Helsinki (to N.D.-P.), Institute of Biotechnology (to N.D.-P.), Biocentrum Helsinki (to N.D.-P.), and Academy of Finland (to N.D.-P.).

## Author contributions

F.O.D.S. and N.D.-P. designed the experimental approach. F.O.D.S., K.M., and N.D.-P. selected the species sampling, and micro-CT scans were carried out by F.O.D.S., K.M., J.O., and J.M. F.O.D.S. collected 2D and 3D landmark data. F.O.D.S., A.-C.F., Y.S., A.H., J.O., and N.D.-P. performed the experiments. F.O.D.S., A.-C.F., Y.S., and N.D.-P. analyzed the data. F.O.D.S. and N.D.-P. collected and prepared some of the reptile embryos. F.O.D.S. and N.D.-P. prepared the figures and wrote the paper and all co-authors contributed in the form of discussion and critical comments. All authors approved the final version of the manuscript.

## Additional information

**Competing interests:** The authors declare no competing financial interests.

