## [Peer Review File · Nature Communications]

Reviewers' comments:

Reviewer #1 (Remarks to the Author):

This is a charming large-scale study of the shape of lizard and snake skulls across multiple biological levels to address the question of the ecological origins of snakes. The major claim of the paper is that the estimated ancestor of the crown-group of snakes is likely to be fossorial, but that of the ancestor to all snakes is estimated as terrestrial. This is a heavily debated topic, where in the past the proponents have fought for an aquatic versus a terrestrial (loosely defined) ancestry of snakes. The authors here are supporting terrestrial (not aquatic) ancestry, which is now the consensus view in the field.

The paper presents an enormous amount of work, which is unique and commendable. The shape methods and statistical analyses are all appropriate (but see convergence below), the manuscript is mostly clear and well-written. The figures are clear and easy to follow. I have a few issues to raise about the ecological classifications, interpretations of the data and terminology used in the manuscript.

My main issue with the manuscript resides in the ecological categories, which is what the main findings are based upon.

The authors present ecological coding in sup figure 5, showing fossorial, burrowing, leaf-litter and semi-fossorial ecologies. How do the authors find a clear-cut difference between fossorial and burrowing? The two definitions seem to overlap – living and foraging underground, and digging for shelter or swimming in sand. Are these based on any data or is it qualitative/speculative? And does this categorisation match what other studies have done? Furthermore, Page 9, line 16 – why does “adapted to a subterranean environment” not correspond to burrowing? Page 9, line 23 – amphisbaenians are dedicated burrowing species, but gymophthalmids are arguably leaf-litter dwellers (they hide under rotten logs, but their long tails suggest they do not actually dig. I am not convinced that these categories are meaningful or lacking subjectivity of the authors.

Terminology: Ecomorphology

Page 2, Line 2 – Ecomorphological hypothesis is incorrect; this is an ecological hypothesis, since only ecology is being discussed. Here and throughout (i.e. P2, line 16, page 12, line 1) this should be checked and changed where appropriate when referring specifically to ecology. Unless the ecology is being defined based on morphology; in which case there is an issue of circular reasoning here.

Terminology: Qualitative versus quantitative.

Through the manuscript, the authors claim novelty and superiority for their “quantitative” approach, brushing aside previous work for its “qualitative” approach. E.g., Page 2, Line 22 – I disagree with the statement that “largely focused on qualitative morphological differences”. There are numerous quantitative studies of squamate morphology at different taxonomic levels and across different structures (see the reference list!). Be specific in what structures you believe to be lacking quantitative analysis, or simply

remove this sweeping and inaccurate statement.

Page 2, Lines 28-30 – by “qualitative” and the references given, I take it you are referring to cladistic coding methods. These are not “lesser” than morphometric methods; they are complementary.

Page 3, Line 4 – “quantitative data” here and throughout is referring specifically to continuous, morphometric data. Need to be explicit here. Because cladistic methods are quantitative too, they are just coding on a discrete, nominal scale.

There is simply no need for this “one method is better than another” prose throughout. Morphometric approaches, using continuous data, are only valid when the same homologous structures are present to measure. Discrete coding approaches are good when characters are present/absent across a sample. These methods are complementary; one is not better than the other.

Following from this, Page 2, Line 4 – need a comma after quantitative; otherwise it’s a tautology (a geometric morphometric approach is quantitative by definition). And page 9, line 8 – again, either put in a comma, or simply remove quantitative.

Sampling.

Many of the specimens studied here are drawings, according to sup Table 1. Called “accurate drawings” in the supplementary methods. How do the authors support the accuracy of this information? And how are the different media types comparable?

What percentage of the total Squamata is sampled here, and how would the morphospace differ with inclusion of more species? The number of species is not mentioned in the main manuscript nor in the supplementary methods), only the number of skulls.

Convergent Evolution.

The authors claim convergent evolution of some lizards and snakes that are “fossorial”, but there are no explicit tests for convergent evolution – instead it is only implied in the phylomorphospace plot. Methods exist to test for convergent evolution under different a priori hypotheses, as outlined by Stayton (2015 Evolution). These should be implemented if this conclusion is to be made.

Ancestral state reconstructions.

ASR my squared-change parsimony (or maximum likelihood) is a weighted averaging process, where ancestral nodes will always be estimated intermediate in morphospace. This caveat should be mentioned in the authors discussion.

Stepping beyond the data.

Page 9, lines 27-29 & Page 10, lines 14-19 – How do the authors make this claim about the head and body evolving together when they only studied cranial shape differences among species and through ontogeny? Page 10, lines 16-17 – How have the authors shown gradual evolution of the unique snake body plan? These data are not in the manuscript.

Minor comments

Page 2, Line 26 – similarly; I disagree with the claim for first and novelty here.

Page 2, Line 35 – add in 'phylogenetic' before comparative methods.

Page 4, line 6 – delete "first"

Page 5, line 10 – No. It suggests convergent evolution. There is not explicit test of convergent evolution being made here (e.g. Stayton 2015 Evolution).

Page 6, line 9 – what does this sentence mean? How does shape and fossoriality overlap?

Page 9, line 9 – what theoretical morphospace? The authors do not do any theoretical modelling here. What they are referring to is the estimated morphospace, based on their measurements.

R Packages: Citation for geomorph (small g), please use CRAN package, as this is more relevant than the MEE paper. geiger is also all small letters. And provide versions for all R packages used.

Supp Figure 6 – what does the k stand for in the centroid size scale? Supp Table 6 shows centroid sizes from 4 to 202,721. So given k is supposedly 1000, yet the scale doesn't make sense given that a large number of the specimens are under 1000 in the table, but not presented as such in Figure 6. And how does this compare to the scale presented in Figure 5?

Supp Table 1 – What does C&S stand for?

Some potentially overlooked references pertaining to ecomorphology in squamates:

Reynolds, R. Graham, et al. "Ecological specialization and morphological diversification in Greater Antillean boas." *Evolution* 70.8 (2016): 1882-1895.

Esquerré, Damien, and J. Scott Keogh. "Parallel selective pressures drive convergent diversification of phenotypes in pythons and boas." *Ecology letters* 19.7 (2016): 800-809.

Hipsley, Christy A., and Johannes Müller. "Developmental dynamics of ecomorphological convergence in a transcontinental lizard radiation." *Evolution* 71.4 (2017): 936-948.

Reviewer #2 (Remarks to the Author):

The main thrust of this paper is morphometric, and I am not a morphometrician. So I cannot comment on the appropriateness of the analytical tools used in this study.

Having said that, the paper comes across as a highly sophisticated confirmation of some long-standing issues in the evolution of squamates, particularly the striking degree of convergence in skull evolution and body elongation in several lineages of burrowing non-ophidian squamates and basal extant snakes, the discussion of which in an ecological and phylogenetic context has a long history. As also has the discussion of scolecophidians as, on the one hand, the most basal clade(s) of extant snakes, yet at the same time characterized by striking specializations of their feeding mechanics. As stressed by the authors on

morphometric grounds, scolecophidians are thus not simply ancestral to alethinophidians. Which brings us to a central issue with this paper, the distinction of ancestors versus sister groups, given the fact that phylogenetic systematics (cladistics) has replaced the search for ancestors by the search for sister group relationships.

The phylogenetic backbone on this paper is a composite phylogeny pulled from several recent comprehensive analyses of squamate phylogeny. The non-ophidian part of this tree reflects the molecular signal, with the highly derived, blind and limbless, burrowing dibamids as the most basal clade, and the iguanians nested within Toxicofera as sister group of anguimorphs. Whereas the molecular support for that topology is very strong, it clashes dramatically with the morphological signal that places the iguanians at the base of the squamate tree (Conrad, 2008; Gauthier et al., 2012). Controversy over these incongruent tree topologies centers on the degree of congruence amongst the molecular data in support of the Toxicofera topology, and on the rate of evolution of the genes employed.

But whereas the Toxicofera topology does not immediately affect the reconstruction of the ancestral snake morphology aimed at by the authors, other aspects of the phylogenetic backbone they employ do so. The scolecophidians are placed as the most basal clade(s) of snakes, a placement that for a long time (at least since Bellairs and Underwood, 1951) has been used to support a fossorial (burrowing) ancestry of snakes – as is also the conclusion of this study based on morphometrics, i.e., that the “crown snake ancestor” was fossorial. That conclusion is based, however, on the phylogenetic placement of the fossil taxa as chosen by the authors, i.e., all nested within “crown snakes”. But there are several phylogenetic analyses that have placed some, or all of these fossil taxa outside the crown snakes, i.e., on the stem of the crown snakes – the latter the extant snakes (with scolecophidians at the base of the crown snakes). It is this placement of fossil snakes on the stem of the crown snakes that has in the last 20 years triggered the debate about a terrestrial, burrowing, or marine origin of snakes. If the fossil snakes are stem-group taxa (rather than crown group taxa as in this paper), then a fossorial origin of snakes is not unambiguously supported on phylogenetic grounds.

The authors chose mosasauroids as sister group of “crown snakes” – in itself another very controversial issue – and labeled the node that links mosasauroids with ‘crown snakes’ “Deep snake” (Fig. 1). With this phylogenetic background comes one of the main conclusions of the paper (e.g., p. 3, lines 27-29): “our data reveal that while the crown-snake ancestor had a small skull with a shape fully adapted to a fossorial lifestyle, all snakes unprecedentedly evolved from a terrestrial form with non-fossorial behavior, thus indicating a new surface-terrestrial-to-fossorial scenario at the origin of snakes.” However, the node in the cladogram labeled “Deep snake” is not the ancestor of “all snakes”, but instead specifies the mosasauroids as sister group of the ‘crown snakes’. However, if – for the sake of the argument – we do place an ancestor at this “Deep snake” node, then the descendants of this ancestor are not “all snakes”, but instead the mosasauroids plus all snakes, i.e., ‘crown snakes’. And that ‘all snakes’ on the one hand, the secondarily marine mosasaurs on the other, would originate from a terrestrial ancestor is again a long held position amongst herpetologists, here again supported by a sophisticated array of statistical tools.

In summary, then, the use of the concepts of “all snakes” and “crown snakes”, each with separate ancestors, is misleading, and even wrong when applied to the phylogenetic backbone presented in Fig. 1.

Snake origins: balance of selection and development as revealed by skull evolution

Filipe O. Da Silva, Anne-Claire Fabre, Yoland Savriama, Joni Ollonen, Kristin Mahlow, Anthony Herrel, Johannes Müller & Nicolas Di-Poï

Point-by-point replies to the reviewers' and editorial comments and suggestions

We thank the two Reviewers for the positive evaluation of our manuscript. We have carefully considered all the comments and recommendations and we explain below how we revised the entire paper to comply with these observations. We reproduce the referees' comments in italics and our responses and explanations are in plain text.

Reviewer #1 (Remarks to the Author):

This is a charming large-scale study of the shape of lizard and snake skulls across multiple biological levels to address the question of the ecological origins of snakes.

We thank the Reviewer for her/his positive evaluation and we provide below the answers to all her/his comments.

The major claim of the paper is that the estimated ancestor of the crown-group of snakes is likely to be fossorial, but that of the ancestor to all snakes is estimated as terrestrial. This is a heavily debated topic, where in the past the proponents have fought for an aquatic versus a terrestrial (loosely defined) ancestry of snakes. The authors here are supporting terrestrial (not aquatic) ancestry, which is now the consensus view in the field.

As highlighted by this expert, the origins of snakes is still a heavily debated topic, thus highlighting the importance of and warranting new complementary, large-scale studies in the field. Conflicting ecological hypotheses for early snakes have been proposed, including aquatic, terrestrial, but also fossorial or even multiple habitats. We agree that a non-aquatic, terrestrial ancestry is currently the "most accepted" scenario, but as also mentioned by this reviewer, the terrestrial habitat has been "loosely defined" so far (e.g., fossorial and leaf-litter species are also terrestrial) and is still debated. Our large-scale and comparative geometric morphometric study helps to clarify such debate by indicating with high confidence an underexplored surface-terrestrial (in the sense not aquatic, fossorial, nor leaf-litter)-to-fossorial transition at the origin of snakes. As suggested by this reviewer (see comment below), we also further revised the ecological categories to better assess such ecological origins of snakes.

The paper presents an enormous amount of work, which is unique and commendable. The shape methods and statistical analyses are all appropriate (but see convergence below), the manuscript is mostly clear and well-written. The figures are clear and easy to follow. I have a few issues to raise about the ecological classifications, interpretations of the data and terminology used in the manuscript.

We thank this reviewer for her/his constructive comments that significantly improved the whole manuscript, including the significance of our conclusions on the fossorial convergence of squamate species and the ecological origins of snakes (see below). We provide below the replies to all her/his

“few” issues. We agreed with most comments and suggestions and have substantially revised the text and some figures, in many respects.

My main issue with the manuscript resides in the ecological categories, which is what the main findings are based upon.

The authors present ecological coding in sup figure 5, showing fossorial, burrowing, leaf-litter and semi-fossorial ecologies. How do the authors find a clear-cut difference between fossorial and burrowing? The two definitions seem to overlap – living and foraging underground, and digging for shelter or swimming in sand.

We agree with this reviewer that some ecological definitions for squamate species can be confusing and/or overlapping - fossorial and burrowing species are both living and foraging underground, even though not necessarily in the same type of ground, and semi-fossorial species can also show some digging or leaf-litter behavior -, including in the published literature. In all cases, we followed descriptions in the literature for the most prominent ecology, which included, in many cases, our own observations in the field (see also comment below). The use of different wildlife observations (with no clear-cut behavior distinction), different habitat definitions, ambiguity in terminology (e.g., fossorial vs burrowing), and different habitat categories for the same species in the previous literature made our large-scale ecological classification challenging. We now carefully rechecked all species sampled in order to simplify the ecological categories by, e.g., revisiting the less-documented semi-fossorial ecology and by merging overlapping categories such as fossorial and burrowing (see also below).

Are these based on any data or is it qualitative/speculative? And does this categorisation match what other studies have done?

As now clearly mentioned in the Supplementary Information material, habitat preferences are not speculative but were gathered from published literature and/or reptile databases such as the IUCN Red List of Threatened Species, the Reptile Database, and the Global Invasive Species Database”. We have now included all references in the new Supplementary Tables 1 and 2 (see 227 new references in Supplementary Information). To limit speculative definition in previous literature and noise due to overlapping of ecologies, habitat preferences were now carefully rechecked and revised for all species sampled based on published literature, reptile databases and/or personal field observations (including by co-authors) as well as discussions with external experts in reptile ecology, and then simplified into five main categories (see revised ecologies and references in the new Supplementary Tables 1 and 2): aquatic (including marine and semi-aquatic species), terrestrial (adapted for surface locomotion and foraging, including saxicolous), leaf-litter (terrestrial but living under vegetation layers or surface debris), fossorial (living and foraging underground), and arboreal (adapted for locomotion between tree branches or bushes, including semi-arboreal).

We also re-performed all ecological analyses using these revised ecological categories, including the different tests of the influence of habitat modes on skull shape, the reconstructions of ancestral ecologies, and a new convergence test (based on reviewer’s comments, see below). These new results are now presented in the revised Figure 4, Supplementary Figure 5, and Supplementary Tables 5-6. Importantly, these new analyses confirmed our previous observations and conclusions on the significant shape differences between some ecological groups, including between the fossorial ecology and all other habitat modes, and reinforced with higher confidence the non-fossorial (>99.9%) but terrestrial (70%) origin of the “most recent common ancestor (MRCA) of snakes and their sister group” (previously named “deep snake ancestor”, see comments of reviewer

2) and the fossorial ecology (82% of likelihood) of the “MRCAs of crown snakes” (previously named “crown snake ancestor”, see comments of reviewer 2). Both the main text (including “Results”, “Discussion” and “Methods” sections) and Supplementary Data 2 on the ecological origins of snakes were modified accordingly.

Furthermore, Page 9, line 16 – why does “adapted to a subterranean environment” not correspond to burrowing?

We agree that the term “burrowing” was often used as synonymous of “fossorial” in the initial manuscript because of the overlapping definitions. We now rechecked and better defined the main habitat ecological categories by, e.g., merging these two overlapping categories (see above). The term “fossorial” is now used throughout the manuscript to indicate life under the surface.

Page 9, line 23 – amphisbaenians are dedicated burrowing species, but gymophthalmids are arguably leaf-litter dwellers (they hide under rotten logs, but their long tails suggest they do not actually dig. I am not convinced that these categories are meaningful or lacking subjectivity of the authors.

We now revised the main habitat preferences for all species (see above). However, as in our original study (see above and Supplementary Tables 1 and 2), our new definitions confirm that all amphisbaenians sampled in our study are fossorial/burrowing species, while gymophthalmids include leaf-litter, terrestrial, and fossorial species (see, e.g., Barros, F.C. *et al.*, *J. Evol. Biol.* **24**, 2423-2433 (2011)). We are convinced that these categories are meaningful as they are based on the extensive ecological literature and field observations (including by co-authors) available for these two squamate groups.

Terminology: Ecomorphology

Page 2, Line 2 – Ecomorphological hypothesis is incorrect; this is an ecological hypothesis, since only ecology is being discussed. Here and throughout (i.e. P2, line 16, page 12, line 1) this should be checked and changed where appropriate when referring specifically to ecology. Unless the ecology is being defined based on morphology; in which case there is an issue of circular reasoning here.

We agree with this reviewer that previous investigations have assessed the “ecological” origins of snakes, so only the ecological hypothesis should be referred when mentioning these studies. However, in our work, we also performed an “ecomorphological” study by assessing the relationship between the skull morphology/shape of snakes and their ecology. In particular, and as mentioned by this reviewer, the ecologies of snake ancestors have been reconstructed based on skull shape parameters. We now checked the whole manuscript (including Supplementary Information) and replaced “ecomorphology” by “ecology” when appropriate, including at positions mentioned by the reviewer.

Terminology: Qualitative versus quantitative.

Through the manuscript, the authors claim novelty and superiority for their “quantitative” approach, brushing aside previous work for its “qualitative” approach. E.g., Page 2, Line 22 – I disagree with the statement that “largely focused on qualitative morphological differences”.

We totally agree with this reviewer that the comparison of “quantitative” vs “qualitative” has been overstated in our initial manuscript. As mentioned by the same reviewer below, “qualitative” was in fact referring to cladistic methods. We now removed the “qualitative” term throughout the manuscript and refer instead to the different methods and/or characters used: “cladistic analysis”,

“discrete traits”. As also suggested by this reviewer (see comment below), we also removed the “quantitative” term when referring to “geometric morphometrics”, as the latter methods are quantitative by definition.

There are numerous quantitative studies of squamate morphology at different taxonomic levels and across different structures (see the reference list!).

We agree that other recent geometric morphometric studies integrating ecological and/or developmental data have revealed new insights into squamate skull evolutionary specializations at different taxonomic levels in several lizard and snake radiations, and we now modified the manuscript accordingly to highlight these studies, including the three publications mentioned below by this reviewer (see below and new references 30-33 in the revised main text). Similarly, we added in the “introduction” section additional recent publications referring to skull heterochrony in squamates at different taxonomic levels (see new references 39-41).

Be specific in what structures you believe to be lacking quantitative analysis, or simply remove this sweeping and inaccurate statement.

A large-scale and integrative comparative geometric morphometric analysis of both skull shape and size has not been performed across the whole of Squamata and in the context of snake origins, neither using embryonic data. We clarified the novelty of our study in the revised manuscript by also citing other morphometric studies recently published within specific clades of lizards or snakes (see above).

Page 2, Lines 28-30 – by “qualitative” and the references given, I take it you are referring to cladistic coding methods. These are not “lesser” than morphometric methods; they are complementary.

We agree with this comment and have modified the text accordingly (see above) to avoid this perception. We also introduced the “complementary” aspect of the different approaches in the “introduction” section of the revised manuscript, by specifically indicating that “The analysis of morphological data using complementary morphometric approaches has the potential to shed light on these issues”.

Page 3, Line 4 – “quantitative data” here and throughout is referring specifically to continuous, morphometric data. Need to be explicit here. Because cladistic methods are quantitative too, they are just coding on a discrete, nominal scale.

We agree with this comment and have modified the whole manuscript accordingly by removing both “qualitative” and “quantitative” terms and by only referring to the different methods and/or characters used (see above).

There is simply no need for this “one method is better than another” prose throughout. Morphometric approaches, using continuous data, are only valid when the same homologous structures are present to measure. Discrete coding approaches are good when characters are present/absent across a sample. These methods are complementary; one is not better than the other.

We agree with this comment and have modified the whole manuscript accordingly by only referring to the different methods and/or characters used (see above). We also now introduced the “complementary” aspect of the different approaches (see above).

Following from this, Page 2, Line 4 – need a comma after quantitative; otherwise it’s a tautology (a geometric morphometric approach is quantitative by definition). And page 9, line 8 – again, either put in a comma, or simply remove quantitative.

As mentioned above, we now removed the “quantitative” term or replaced it by “comparative” throughout the manuscript when referring to “geometric morphometrics”.

Sampling.

Many of the specimens studied here are drawings, according to sup Table 1. Called “accurate drawings” in the supplementary methods. How do the authors support the accuracy of this information? And how are the different media types comparable?

This expert is right in raising this issue about accuracy of drawings, and this is precisely the main reason why only “accurate drawings” made by anatomists and taxonomists and published in highly respected publications and/or specialized books on squamate anatomy (see references in Supplementary Information material) have been used in our study. We now better defined “accurate drawings” in the revised Supplementary Data 1.

The different media types are comparable in our 2D analyses because of the standard lateral orientation traditionally used to report squamate specimens in descriptive works (see references in Supplementary Tables 1 and 2). Importantly, for the consistency of the different sources of 2D data (including accurate drawings), we tested for the correct, similar lateral positioning of the skull by incorporating whenever possible more than one specimen or more than one reconstruction or original skull picture per species (including for fossils), thus circumventing imaging problems and/or misinterpretations in skull data. Both the accuracy and correct positioning of skulls in lateral view were further improved by controlling for shape outliers in the software MorphoJ v1.06, and by comparing different sources of data (including 2D and 3D skull information) for the same species and/or genus when available (see above). In a very few species, we observed unexpected outliers because of skull positioning; these skull pictures were either excluded or recaptured after repositioning of the specimens in lateral view, thus improving the quality of our data. Finally, because of bone movements in open and closed mouth positions, only species with closed mouths were selected. Most importantly, we show here that the results and conclusions obtained from both 2D and 3D data converge (see Figure 3 and Supplementary Figure 3). These quality and congruence checks ensured the quality of our datasets for morphometric investigations. We now introduced more information about the 2D data sources and quality checks in the “Methods” section of the revised main text, and added full details in the revised Supplementary Data 1.

What percentage of the total Squamata is sampled here,

The number of extant squamate species sampled represents approximately 3% of total Squamata, based on the total number of known extant species reported in the August 15th, 2016 version of the Reptile Database (<http://www.reptile-database.org>). This dataset allowed us to perform the first large-scale synthesis of skull shape diversity in the whole of Squamata, by covering all lineages and most families of squamates (only 6 families could not be sampled because of scarce literature or rarity of samples in museum collections). We now mentioned this % in the “Methods” section of the revised main text as well as in the Supplementary Information material.
and how would the morphospace differ with inclusion of more species?

As already mentioned in the first manuscript, both 2D and 3D morphometric analyses as well as several alternative composite phylogenetic hypotheses and different recent combined molecular and

morphological phylogenetic trees were used to increase the robustness of our study. This allowed us to address the phylogenetic uncertainty of some fossils but also to test impact of branch lengths and species number. Importantly, we show here that the results and conclusions obtained from 2D (326 species) and 3D (100 species) data or from different phylogenies (326 species in our main composite phylogeny hypothesis, 277 species in phylogeny of Pyron *et al.* 2013, 147 species in phylogeny of Reeder *et al.* 2015, and 60 species in phylogeny of Hsiang *et al.* 2015) converge, and that the inclusion of additional species only fills some “gaps” in the morphospace (see, e.g. Figure 3 and Supplementary Figure 3). We now inserted more information about these different tests (including different species numbers which was not mentioned in the first manuscript) in the revised main text (based also on the comments of reviewer 2, see below), and added full details in the revised Supplementary Data 1.

The number of species is not mentioned in the main manuscript (nor in the supplementary methods), only the number of skulls.

This is an understandable request. In addition to the number of skulls, we now indicate the number of species sampled for adult, embryonic and fossil taxa (408 skulls in total corresponding to 326 species) in the revised main text (see “Results” and “Methods” sections) and Supplementary Information material. We also included the number of embryonic species used in our ontogenetic analyses in the revised main text.

Convergent Evolution.

The authors claim convergent evolution of some lizards and snakes that are “fossorial”, but there are no explicit tests for convergent evolution – instead it is only implied in the phylomorphospace plot. Methods exist to test for convergent evolution under different a priori hypotheses, as outlined by Stayton (2015 Evolution). These should be implemented if this conclusion is to be made.

As proposed by the reviewer, we now quantified convergent evolution based on the new ecological categories (see above), using computation of the distance-based convergence measures C1-C4 in the R-package *convevol* v1.1, as described in the publication of Stayton (*Evolution* 69, 2015). Significance was assessed in the same package using 1000 evolutionary simulations along the phylogeny according to a Brownian motion-model, as described by Stayton. These data are now presented in the new Supplementary Table 5 and in the “Results” section of the main text on the “ecological origins of snakes”. We thank the reviewer for this suggestion as these analyses confirm the significant convergence of all fossorial snake and lizard species (Supplementary Table 5), as previously suggested by the phylogenetic trends in our phylomorphospace plot.

Ancestral state reconstructions.

ASR my squared-change parsimony (or maximum likelihood) is a weighted averaging process, where ancestral nodes will always be estimated intermediate in morphospace. This caveat should be mentioned in the authors discussion.

As mentioned in the Supplementary Information and revised main text (“Methods” and “results” sections), we used both unweighted and weighted squared-change parsimony algorithms and different phylogenetic hypotheses to estimate skull shape and size evolution. Weighted squared-change parsimony estimates are equivalent to maximum likelihood estimates under Brownian motion, and we agree with the reviewer that ancestral nodes will always be estimated intermediate using these ancestral character-state reconstruction methods. We now replaced “modelled” by “estimated” throughout the manuscript when referring to ancestral reconstructions, and we rather

mentioned the limitations of the methods in the “Methods” section of the revised main text, as all our reconstructions and conclusions converge and indicate a lizard-to-snake transition proceeding along the PC2 axis in the phylomorphospace.

Stepping beyond the data.

Page 9, lines 27-29 & Page 10, lines 14-19 – How do the authors make this claim about the head and body evolving together when they only studied cranial shape differences among species and through ontogeny?

The parallels between cranial (our studies) and body (previous studies) evolution was in fact only one hypothesis related to the previously proposed body-first and head-first hypotheses on snake evolution. We now modified the second paragraph in the “discussion” section of the main text to better emphasize the hypothetical aspect of this comparison, e.g., by indicating that the body aspect was “previously reported in snakes” and that “skull and body may have evolved jointly in snakes”. We agree that this developmental aspect goes beyond the data and we now completely removed this hypothesis from both the “abstract” and last paragraph of the “discussion” of the main text. However, this is a very interesting issue, for which we unfortunately have only speculations, and we indicate in the “discussion” section that “similar morphometric and ontogenetic analyses of axial anatomy would be needed to confirm this hypothesis”.

Page 10, lines 16-17 – How have the authors shown gradual evolution of the unique snake body plan? These data are not in the manuscript.

As mentioned above, we agree that the claim on the head-body relationship goes beyond the data and we now completely removed the “body” aspect from the last paragraph of the “discussion” of the main text, by focusing on the major findings of our study on skull evolution. We also removed “gradual” from the sentence to avoid any mis-interpretation.

Minor comments

Page 2, Line 26 – similarly; I disagree with the claim for first and novelty here.

As mentioned above, a large-scale and integrative comparative geometric morphometric analysis of both skull shape and size throughout ontogeny had not been performed across the whole of Squamata and in the context of snake origins. We revised this sentence by removing “the first” and by indicating that “we performed a large-scale and integrative geometric morphometric analysis of skull bones across squamates to help clarify the ecological and evolutionary origins of snakes”. We also modified a similar sentence in the last paragraph of the “Introduction” section. We also clarified the novelty of our study in the context of snake origins throughout the revised manuscript, and by also citing other morphometric studies recently published at different taxonomic levels within specific clades of lizards or snakes (see above).

Page 2, Line 35 – add in ‘phylogenetic’ before comparative methods.

This has been corrected.

Page 4, line 6 – delete “first”

This has been deleted.

Page 5, line 10 – No. It suggests convergent evolution. There is not explicit test of convergent evolution being made here (e.g. Stayton 2015 Evolution).

We agree that our phylomorphospace plot only suggests convergent evolution, and we now replaced “confirms” by “suggests” in this particular sentence. We also modified “confirming” by “implying” in the same paragraph when referring to convergent evolution based on phylogenetic signal estimation. In addition, as suggested by this reviewer (see above), we also now quantified and assessed the significance of convergent evolution in our revised ecological analysis. This convergence test is now mentioned in the “Results” section on the “ecological origins of snakes” and significantly support the convergence of fossorial lizard and snake species (see above).

Page 6, line 9 – what does this sentence mean? How does shape and fossoriality overlap?

We now clarified this sentence by simply indicating that “a discrete area containing all fossorial lizard and snake species clearly separates from other ecologies at negative PC2 values”.

Page 9, line 9 – what theoretical morphospace? The authors do not do any theoretical modelling here. What they are referring to is the estimated morphospace, based on their measurements.

This has been corrected by replacing “theoretical” by “estimated”.

R Packages: Citation for geomorph (small g), please use CRAN package, as this is more relevant than the MEE paper.

We now replaced “Geomorph” by “geomorph” throughout the manuscript and cite both the publication of Adams *et al.* 2013 and the CRAN package repository website in the main manuscript (“Results” section) as well as in the Supplementary Information.

geiger is also all small letters. And provide versions for all R packages used.

We now modified “GEIGER” by “geiger” throughout the manuscript and provide all versions of the packages (MorphoJ, geomorph, Geiger, MASS, convevol) and softwares (Amira, tpsDig) used in our study in both the main text and Supplementary Information.

Supp Figure 6 – what does the k stand for in the centroid size scale? Supp Table 6 shows centroid sizes from 4 to 202,721. So given k is supposedly 1000, yet the scale doesn’t make sense given that a large number of the specimens are under 1000 in the table, but not presented as such in Figure 6.

We think that this expert has accidentally merged the two Supplementary tables showing the centroid size values. Indeed, centroid size values are listed in two different tables, one for the 2D data (new Supplementary Table 7; values from 4 to 219, excluding the gigantic mosasaurs) and one for the 3D data (new Supplementary Table 8, values from 9,551 to 202,721), and both raw values (4th column) and log-transformed values (5th column) are given. Supplementary Figure 6 shows the raw values of the centroid size for the 3D data (according to the new Supplementary Table 8), thus explaining why all values are above 9,000. It is worth noticing that the centroid size is calculated based on the disposition of landmarks, and 3D data has more landmarks. We hope that these details will help the reviewer in his new examination of our data and figures.

And how does this compare to the scale presented in Figure 5?

Similarly, Figure 5 shows the raw values of the centroid size for the 2D data (according to the new Supplementary Table 7), with values under 250 because of the absence of mosasaur genera, as indicated in the figure legend (“mosasaur genera are not shown because of their gigantic size”). Figure 6 shows the log-transformed values of the centroid size for the 2D data (according to the new Supplementary Table 7), with values between 1 and 6 for extant species.

Supp Table 1 – What does C&S stand for?

“C&S” stands for “cleared and alcian blue/alizarin red-stained”. We now defined this abbreviation in the Supplementary Data 1 and legend of Supplementary Table 1.

Some potentially overlooked references pertaining to ecomorphology in squamates:

Reynolds, R. Graham, et al. "Ecological specialization and morphological diversification in Greater Antillean boas." Evolution 70.8 (2016): 1882-1895.

Esquerré, Damien, and J. Scott Keogh. "Parallel selective pressures drive convergent diversification of phenotypes in pythons and boas." Ecology letters 19.7 (2016): 800-809.

Hipsley, Christy A., and Johannes Müller. "Developmental dynamics of ecomorphological convergence in a transcontinental lizard radiation." Evolution 71.4 (2017): 936-948.

As mentioned above, these three references (among other studies) are now cited in the revised manuscript to highlight previous morphometric investigations performed at different taxonomic levels in squamates. We really thank again this reviewer for her/his constructive comments that significantly improved the whole manuscript.

Reviewer #2 (Remarks to the Author):

The main thrust of this paper is morphometric, and I am not a morphometrician. So I cannot comment on the appropriateness of the analytical tools used in this study. Having said that, the paper comes across as a highly sophisticated confirmation of some long-standing issues in the evolution of squamates, particularly the striking degree of convergence in skull evolution and body elongation in several lineages of burrowing non-ophidian squamates and basal extant snakes, the discussion of which in an ecological and phylogenetic context has a long history. As also has the discussion of scolecophidians as, on the one hand, the most basal clade(s) of extant snakes, yet at the same time characterized by striking specializations of their feeding mechanics. As stressed by the authors on morphometric grounds, scolecophidians are thus not simply ancestral to alethinophidians.

We thank this expert for highlighting the importance of our new large-scale morphometric study in helping to clarify such long-standing evolutionary issues on snake origins and diversification. We provide below the answers to all her/his comments.

Which brings us to a central issue with this paper, the distinction of ancestors versus sister groups, given the fact that phylogenetic systematics (cladistics) has replaced the search for ancestors by the search for sister group relationships.

We agree with this comment and have now replaced the term “ancestor” by “most recent common ancestor (MRCA)” throughout the manuscript, including in main Figures 4-5 and Supplementary Information material (see also below). In particular, as further commented below by this reviewer,

we modified the names for the two key internal nodes relative to the origin and diversification of snakes, by now referring to “MRCA of crown snakes” (instead of “crown snake ancestor”) and “MRCA of snakes and their sister group” (instead of “deep snake ancestor”).

The phylogenetic backbone on this paper is a composite phylogeny pulled from several recent comprehensive analyses of squamate phylogeny. The non-ophidian part of this tree reflects the molecular signal, with the highly derived, blind and limbless, burrowing dibamids as the most basal clade, and the iguanians nested within Toxicofera as sister group of anguimorphs. Whereas the molecular support for that topology is very strong, it clashes dramatically with the morphological signal that places the iguanians at the base of the squamate tree (Conrad, 2008; Gauthier et al., 2012). Controversy over these incongruent tree topologies centers on the degree of congruence amongst the molecular data in support of the Toxicofera topology, and on the rate of evolution of the genes employed.

As mentioned in the Supplementary Information (and further highlighted now in the main text of the revised manuscript), the phylogenetic backbone of our composite hypothesis was not only set based on molecular phylogenetic topologies, but also on the most recent and inclusive combined molecular and morphological phylogenetic analyses containing the highest number of both phenotypic and molecular characters for squamates. These recent squamate phylogenies (e.g., Hsiang et al. 2015; Reeder et al. 2015; Pyron et al. 2016) have been precisely released to help resolving the conflicts between molecular and morphological phylogenetic hypotheses, by expanding and integrating both morphological and molecular datasets. While large-scale molecular (Pyron et al. 2013; Tonini et al. 2016) and combined molecular and morphological phylogenies largely converge for extant species (including Toxicofera), they differ for the positions of some snake fossils, thus explaining why we tested all these different combined phylogenies in our studies in addition to our main composite hypothesis (see also comment below and Supplementary Information). We did not favor any phylogenies in our studies, but rather tested different hypotheses based on the most recent and inclusive phylogenies available for squamates. We now clarified this key phylogenetic aspect at different positions within the main text (see “Results” and “Methods” sections as well as legend of Figure 1) and Supplementary Information material.

But whereas the Toxicofera topology does not immediately affect the reconstruction of the ancestral snake morphology aimed at by the authors, other aspects of the phylogenetic backbone they employ do so.

As mentioned above, we already accounted for topology variations by using several alternative composite hypotheses as well as different recent combined molecular and morphological phylogenetic trees in our studies (see also below as well as Supplementary Information and revised main text).

The scolecophidians are placed as the most basal clade(s) of snakes, a placement that for a long time (at least since Bellairs and Underwood, 1951) has been used to support a fossorial (burrowing) ancestry of snakes – as is also the conclusion of this study based on morphometrics, i.e., that the that the “crown snake ancestor” was fossorial.

As described above, we did not favor any phylogenies in our studies, and the position of scolecophidians was set based on the most recent and inclusive molecular as well as combined molecular and morphological phylogenetic analyses available so far for squamates (Pyron et al. 2013; Hsiang et al. 2015; Reeder et al. 2015; Pyron et al. 2016; Tonini et al. 2016). Our

conclusions based on morphometrics and ancestral reconstructions indeed indicate with high confidence a fossorial origin for the “MRCA of crown snakes” (previously named “crown snake ancestor”, see above), but not as specialized in skull morphology as scolecophidians (see “Results” section of main text), as also supported by other recent studies (Longrich *et al.* 2012; Li & Norell 2015).

That conclusion is based, however, on the phylogenetic placement of the fossil taxa as chosen by the authors, i.e., all nested within “crown snakes”. But there are several phylogenetic analyses that have place some, or all of these fossil taxa outside the crown snakes, i.e., on the stem of the crown snakes – the latter the extant snakes (with scolecophidians at the base of the crown snakes). It is this placement of fossil snakes on the stem of the crown snakes that has in the last 20 years triggered the debate about a terrestrial, burrowing, or marine origin of snakes. If the fossil snakes are stem-group taxa (rather than crown group taxa as in this paper), then a fossorial origin of snakes is not unambiguously supported on phylogenetic grounds.

Our conclusions are not based on the phylogenetic placement of fossil data but on ancestral reconstructions considering the entire large-scale dataset. As mentioned in the Supplementary Information (and now in the main text of the revised manuscript, see also above), several alternative composite hypotheses as well as different recent combined molecular and morphological phylogenetic trees were also used in our morphometric studies, to specifically assess the impact of 2D and 3D data, species number (see also comment of Reviewer 1), branch length information and total absence of fossils, but also to address the phylogenetic uncertainty of some fossil taxa. In particular, and as mentioned by this reviewer, some putative stem snake fossils used here such as *Dinilysia* are remarkably unstable in their position across recent combined molecular and morphological phylogenetic trees (Hsiang *et al.* 2015; Reeder *et al.* 2015; Pyron *et al.* 2016), so we also tested different positions for this fossil (as crown snake or sister-taxon to all snakes). Importantly, very similar morphospaces and skull shape predictions for the “MRCA of crown snakes” were obtained for all phylogenies tested, thus ensuring the robustness of our results and conclusions (see e.g., Figures 3 and Supplementary Figure 3). We now further detailed these key points at different positions within the main text (see “Results” sections) and Supplementary Information material. As mentioned in the main text and Supplementary Information, it is worth noticing that the skull of other putative stem snakes like *Coniophis*, *Tetrapodophis*, and *Najash* were too incomplete to be used in our study.

The authors chose mosasauroids as sister group of “crown snakes” – in itself another very controversial issue –

As mentioned above, we did not favor any phylogenies in our studies, and the position of mosasauroids was set based on the most recent and inclusive combined molecular and morphological phylogenetic analyses including these fossils (Reeder *et al.* 2015; Pyron *et al.* 2016). We further tested different phylogenetic hypotheses with and without mosasauroids (see Supplementary Information), as these species are, e.g., not included in our 3D analysis nor in our analysis using the phylogeny of Hsiang *et al.* 2015. Similar morphospaces and ancestral skull shape predictions were obtained in these tests (see e.g., Figures 3 and Supplementary Figure 3). Finally, we further estimated the ecology of the MRCA of Toxicofera in our main composite hypothesis, and also confirm a terrestrial, non-fossorial form. We now further detailed this key point at different positions within the revised main text (see “Results” section) and Supplementary Information.

and labeled the node that links mosasauroids with ‘crown snakes’ “Deep snake” (Fig. 1). With this phylogenetic background comes one of the main conclusions of the paper (e.g., p. 3, lines 27-29): “our data reveal that while the crown-snake ancestor had a small skull with a shape fully adapted to a fossorial lifestyle, all snakes unprecedentedly evolved from a terrestrial form with non-fossorial behavior, thus indicating a new surface-terrestrial-to-fossorial scenario at the origin of snakes.” However, the node in the cladogram labeled “Deep snake” is not the ancestor of “all snakes”, but instead specifies the mosasauroids as sister group of the ‘crown snakes’. However, if – for the sake of the argument – we do place an ancestor at this “Deep snake” node, then the descendants of this ancestor are not “all snakes”, but instead the mosasauroids plus all snakes, i.e., ‘crown snakes’.

We agree with this comment and have now replaced the terms “deep snake ancestor” and ‘crown snake ancestor’ by “most recent common ancestor (MRCA) of snakes and their sister group” and “MRCA of crown snakes”, respectively, throughout the manuscript, including in the main Figures 4-5 and Supplementary Information material (see also below). We also removed these ancestors from the internal nodes in Figures 1 and 3. In addition, we corrected the sentence in the main text by indicating that “our data reveal that while the most recent common ancestor (MRCA) of crown snakes had a small skull with a shape fully adapted to a fossorial lifestyle, all snakes plus sister group unprecedentedly evolved from a terrestrial form with non-fossorial and non-leaf-litter behaviors”. Finally, we also corrected a similar sentence referring to the “MRCAs” and “all snakes” in the Abstract.

And that ‘all snakes’ on the one hand, the secondarily marine mosasaurs on the other, would originate from a terrestrial ancestor is again a long held position amongst herpetologists, here again supported by a sophisticated array of statistical tools.

As also highlighted by the reviewer 1, the terrestrial habitat has been “loosely defined” so far (e.g., fossorial and leaf-litter species are also terrestrial) and is still highly debated. Our large-scale and comparative geometric morphometric study helps to clarify such debate and indicates with high confidence a surface-terrestrial (in the sense not aquatic, fossorial, nor leaf-litter) origin for the MRCA of snakes and their sister group (mosasaurs). We also replaced the term “new” by “underexplored” ecological scenario in the abstract to avoid any conflict of novelty about the “terrestrial” origin.

In summary, then, the use of the concepts of “all snakes” and “crown snakes”, each with separate ancestors, is misleading, and even wrong when applied to the phylogenetic backbone presented in Fig. 1.

We totally agree with this comment and have now replaced these misleading terms throughout the manuscript (see above). We really thank again this reviewer for her/his constructive comments that significantly improved the whole manuscript.

REVIEWERS' COMMENTS:

Reviewer #1 (Remarks to the Author):

The authors have made a full revision of their manuscript and incorporated all of the two reviewers' comments with grace, and the resulting manuscript is stronger and enjoyable to read. I also believe it will be controversial and add another dimension to the ongoing debate of snake origins, thus being an important contribution to the literature. The additional analyses of convergent evolution really enhance the manuscript, and I thank the authors for diligently checking their use of terminology throughout.

I have no further issues with the manuscript, and only ask that the authors cite the most recent addition to this literature that has been published in RSOS By Palci et al. "The morphology of the inner ear of squamate reptiles and its bearing on the origin of snakes", and they may wish to comment on this on page 6 where the *Dinilysia* fossil is discussed along with others.

Reviewer #2 (Remarks to the Author):

This is an elegant morphometric study addressing the problem of snake origins. I am not a morphometrician, and hence cannot comment on the main thrust of the paper. But the second reviewer has found the morphometric analyses adequate and commendable. I found that the authors have adequately responded to the reviewers' comments, and have extensively revised their manuscript. As it now stands, it is from my point of view acceptable for publication.

Point-by-point replies to the reviewers' comments and suggestions

We thank the two Reviewers for the positive evaluation of our manuscript. We have carefully considered all the new comments and recommendations. We reproduce the referees' comments in italics and our responses and explanations are in plain text.

Reviewer #1 (Remarks to the Author):

The authors have made a full revision of their manuscript and incorporated all of the two reviewers' comments with grace, and the resulting manuscript is stronger and enjoyable to read. I also believe it will be controversial and add another dimension to the ongoing debate of snake origins, thus being an important contribution to the literature. The additional analyses of convergent evolution really enhance the manuscript, and I thank the authors for diligently checking their use of terminology throughout.

We thank the Reviewer for her/his positive evaluation and we provide below the answer to her/his additional comment.

*I have no further issues with the manuscript, and only ask that the authors cite the most recent addition to this literature that has been published in RSOS By Palci et al. "The morphology of the inner ear of squamate reptiles and its bearing on the origin of snakes", and they may wish to comment on this on page 6 where the *Dinilysia* fossil is discussed along with others.*

This new reference is now cited in the revised manuscript (see new reference 30) to highlight this recent morphometric investigation in the context of snake origins. As proposed by this reviewer, we also now cite this reference in the "Results" section of the revised manuscript. We really thank again this reviewer for her/his constructive comments that significantly improved the whole manuscript.

Reviewer #2 (Remarks to the Author):

This is an elegant morphometric study addressing the problem of snake origins. I am not a morphometrician, and hence cannot comment on the main thrust of the paper. But the second reviewer has found the morphometric analyses adequate and commendable. I found that the authors have adequately responded to the reviewers' comments, and have extensively revised their manuscript. As it now stands, it is from my point of view acceptable for publication.

We really thank again this reviewer for her/his constructive comments that significantly improved the whole manuscript.